# Early Renal Ultrasound in Patients with Congenital Solitary Kidney Can Guide Follow-Up Strategy Reducing Costs While Keeping Long-Term Prognostic Information

**DOI:** 10.3390/jcm11041052

**Published:** 2022-02-17

**Authors:** Stefano Guarino, Anna Di Sessa, Simona Riccio, Daniela Capalbo, Alfonso Reginelli, Salvatore Cappabianca, Pier Francesco Rambaldi, Emanuele Miraglia del Giudice, Cesare Polito, Pierluigi Marzuillo

**Affiliations:** 1Department of Woman, Child and of General and Specialized Surgery, Università degli Studi della Campania “Luigi Vanvitelli”, Via Luigi De Crecchio 2, 80138 Naples, Italy; stefano.guarino@policliniconapoli.it (S.G.); anna.disessa@unicampania.it (A.D.S.); simonariccio92@gmail.com (S.R.); daniela.capalbo@hotmail.it (D.C.); emanuele.miraglia@unicampania.it (E.M.d.G.); cesare.polito@golfonet.it (C.P.); 2Department of Precision Medicine, Università degli Studi della Campania “Luigi Vanvitelli”, 80138 Naples, Italy; alfonsoreginelli@hotmail.com (A.R.); salvatorecappabianca@gmail.com (S.C.); 3Nuclear Medicine, Department of Radiological Sciences, Università degli Studi della Campania “Luigi Vanvitelli”, 80138 Naples, Italy; pierfrancesco.rambaldi@unicampania.it

**Keywords:** solitary kidney, diagnostic ultrasound, outcome assessment, kidney function tests

## Abstract

We aimed to evaluate the prognostic value of renal length (RL) > 2 standard deviation scores (SDS) measured by renal ultrasound (RUS), across infancy, childhood and adolescence, in identifying which patients with congenital solitary functioning kidney (CSFK) are at lower risk of developing kidney injury (KI). We also estimated the cost saving of integrating the current follow-up protocols with an early RUS algorithm (ERUSA). Fifty-six CSFK adult patients who were 1–3 months old at first observation of undergoing RUS were enrolled. KI was defined by hypertension and/or proteinuria and/or declined renal function. ERUSA was assessed by early (at 1–3 months of life) RUS and was retrospectively tested in our patients. ERUSA establishes that patients with RL > 2SDS at early RUS do not undergo further follow-ups. The others undergo another RUS at 1 year of age along with follow-ups according with current protocols, with the exception of RUS which could be no longer performed. Direct and indirect costs were calculated for each analysed protocol and the cost saving of applying ERUSA was calculated. None of the patients with early RL > 2SDS presented KI in adulthood. A RL > 2SDS was predictive of absence of KI only at 1–3 months (OR = infinity) and 1 year of age (OR = 0.13; 95%CI: 0.03–0.66; *p* = 0.01). ERUSA provided a total cost-sparing ranging from 38.6% to 55.3% among the analysed follow-up protocols. With ERUSA, no patients developing KI in adulthood were missed. In conclusion, only a RL > 2SDS at 1–3 months and 1 year of age predicted good prognosis in young adulthood. ERUSA can guide a cost-sparing follow-up strategy in CSFK patients while maintaining important long-term information.

## 1. Introduction

The compensatory growth of a congenital solitary functioning kidney (CSFK) is defined by a renal length (RL) reaching 2 standard deviations (SDS) and is considered a pivotal marker to predict normal kidney function [1,2].

As further confirmation of the prognostic relevance of a RL > 2SDS in CSFK patients, we previously found that a RL > 2SDS in the neonatal period could allow the identification of patients at lower risk of both presenting with vesico-ureteral reflux, itself associated with development of negative outcomes in CSFK patients [3], and future decline of estimated glomerular filtration rate (eGFR) [4].

In fact, a RL > 2SDS in the early post-natal period could reflect renal hyperplasia characterized by increased nephron endowment [5,6,7] and subsequent reduced risk of hyperfiltration and kidney injury (KI) [8].

Most CSFK patients, however, develop renal compensatory hypertrophy (RL > 2SDS) within the first years of life [9]. This does not allow further discrimination of patients born with renal hyperplasia from those having developed only compensatory hypertrophy later in life.

We hypothesized that the prognostic utility of renal ultrasound (RUS) in CSFK reduces with the increase of the age at RUS execution. Therefore, we evaluated the utility of RL > 2SDS, periodically measured across infancy, childhood and adolescence, in selecting patients at lower risk of developing KI in adulthood (primary outcome). To identify early-in-life predictors of absence of KI in adulthood could be useful in personalizing follow-up protocols of CSFK children in order to reduce unnecessary medicalization with impact on stressful procedures (such as blood sample collection in children) and economic costs. We also estimated the cost saving of integrating the current follow-up protocols [2,10] with the early RUS algorithm (ERUSA) (secondary outcome). The ERUSA is based on the hypothesis that early RUS could provide the instruments to personalize follow-up.

## 2. Materials and Methods

This is an analysis of data from a previous study evaluating CSFK patients followed-up in the Pediatric Department of AOU Università degli Studi della Campania “Luigi Vanvitelli” between January 1993 to January 2020, aged 1–3 months at first observation, and having reached adulthood at last observation [11]. The patients were called back for a careful evaluation of kidney function at age 21.1 ± 3.4 years. The study obtained ethical approval from the Institutional Review Board of Università degli Studi della Campania “Luigi Vanvitelli”.

### 2.1. Inclusion and Exclusion Criteria

Inclusion criteria were (i) prenatal diagnosis of solitary kidney (due to unilateral renal agenesis or multi-cystic dysplastic kidney), (ii) first evaluation between 1 and 3 months of life with post-natal confirmation of the CSFK by abdomen ultrasound and Tc99mMAG3 or Tc99mDMSA scintigraphy and check of possible congenital anomalies of the kidney and urinary tract (CAKUT) of CSFK by RUS, voiding cystourethrogram (in males) or cysto-scintigraphy (in females), (iii) being 18 years or older for the follow-up assessment [11]. A consort diagram describing patients’ enrolment and detailing the exclusion criteria is shown in the Figure 1. Among the patients excluded because of postnatal CSFK diagnosis, one of female gender was syndromic. She presented with the Rubinstein-Taybi syndrome and with other comorbidities such as growth hormone deficiency, Arnold Chiari malformation and pituitary hypoplasia [12]. Patients with KI at first observation were excluded from the study because it would be hard to retrospectively establish whether signs of KI early in life were due to acute neonatal complications or if they were an evolution of CSFK.

### 2.2. CAKUTs Definitions

The CSFK was confirmed by abdomen ultrasound and Tc99mMAG3 or Tc99mDMSA scintigraphy [13]. The diagnosis of multi-cystic dysplastic kidney was made in case of a non-functioning kidney at scintigraphy with ultrasound evidence of parenchyma completely substituted by large noncommunicating cysts [13].

Possible CAKUT of CSFK was checked in all subjects by renal ultrasound, voiding cystourethrogram (in males) or cysto-scintigraphy (in females) [13].

We considered the hydronephrosis as expression of ureteropelvic junction obstruction (UPJO) if requiring surgical intervention because of compatible Tc99mMAG3 scintigraphy and anterior-posterior diameter of the renal pelvis > 30 mm at renal ultrasound. Otherwise, the hydronephrosis was defined as non-obstructive (functional) and therefore irrelevant [13].

Primary megaureter was defined by the evidence of any ureteral dilatation > 7 mm [13].

If the hydronephrosis and/or megaureter were associated with vesicoureteral reflux (VUR) the patients were considered as affected only by VUR [13].

Clustering of VUR grades was adapted, classifying Grades III-V as “dilated” and Grades I-II as “non-dilated” [13].

All the patients with dilated VUR or megaureter started antibiotic prophylaxis just after the diagnosis and this was continued up to the first year of life in males and up to the second year of life in females [13].

All the RUS over the years were made by the same group of radiologists (A.R. and S.C.) or by a pediatric nephrologist with particular experience in the RUS (C.P.). All colleagues had at least five-year experience at the first RUS, with an increase in experience over the years. All nuclear medicine exams were performed by the same specialist in nuclear medicine (P.F.R.), director of this service over the last three decades.

### 2.3. Follow-Up Assessment

The patients attended our observation yearly until 5 years of age and then every two years and, if presenting KI, yearly after the KI detection. At every follow-up visit, weight, height, eGFR, renal length, systolic (SBP) and diastolic blood pressure (DBP), urinalysis, and urinary protein/creatinine ratio (UPr/Cr) were evaluated as previously described [13]. In childhood, the creatinine was measured by Jaffé method and the eGFR was calculated by the original Schwartz formula [14].

When the patients reached adulthood, we also evaluated blood pressure by 24 h-ambulatory blood pressure monitoring (ABPM) and urinary albumin. Only at the last follow-up during adulthood, corresponding to the enrolment in the present study, was serum creatinine evaluated by isotope dilution mass spectrometry, and eGFR calculated by CKD-EPI creatinine equation [15].

### 2.4. Kidney Injury Definition

KI was defined by persistent (>3 months) hypertension and/or proteinuria and/or reduced eGFR [11]:−24 h-ABPM confirmed hypertension according to criteria by Williams et al. [16] or SBP and/or DBP > 95th percentile corrected for age, gender and height [17].−UPr/Cr > 0.2 mg/mg or albuminuria/creatininuria ratio (Ua/Cr) > 30 µg/mg [11].−eGFR < 90 mL/min/1.73 m^2^ for children aged >2 years and according to the specific reference values for age for children <2 years [18].

### 2.5. ERUSA

The ERUSA was developed after having analysed the evidence deriving from the primary outcome of this study. This algorithm is shown in the Figure 2. In this algorithm we do not suggest further follow-up evaluations until adulthood for patients with RL > 2SDS at 1–3 months of life in the absence of complex features, with the exception of the controls which children routinely undergo, including blood pressure measurement [19]. For the other patients, ERUSA establishes performing RUS at 1 year and suggests following-up the patients independently from the RL at this age. Moreover, after the RUS at 1 year of age, ERUSA establishes a procedure of no longer submitting patients to RUS in case of absence of complex features. This algorithm could be applied to any available protocol in order to follow-up CSFK patients.

### 2.6. Cost Analysis

The direct costs that would have been incurred for the different follow-up protocols [2,10] were calculated. The costing consisted of the reimbursement of Italian Health System: RUS (€82) [20], blood sample collection (€2.58), creatinine dosage (€1.13), urinalysis (€2.17), follow-up visit (€20.66) [21].

We did not analyze the costs of voiding cystourethrogram or cysto-scintigraphy and kidney scintigraphy because these were no more routinely performed in CSFK patients [2].

The indirect costs for each ordered procedure were calculated. We estimated the time spent on the ordered procedures as follows: 3 h for RUS, creatinine dosage, urinalysis and follow-up visit with blood pressure measurement; 2 h for creatinine dosage and/or urinalysis and follow-up visit with blood pressure measurement; 1 h for follow-up visit with blood pressure measurement. We calculated the indirect costs on the basis of a mean hourly gross wage of €17.29 in Italy multiplied for the time spent on ordered procedures as detailed above [22]. We included an estimation of €15 for transportation costs for each follow-up visit with the concomitant ordered procedures.

The cost analysis was made accounting for all procedures ordered (RUS, serum creatinine measurement, urinalysis, and follow-up visit with blood pressure measurement) within the first 18 years of age.

We estimated the costs sustained according with the old approach adopted in our clinic, with the approach suggested by Groen in ‘t Woud et al. [2] and that suggested by Jawa et al. [10], as well as the cost saving created by applying ERUSA to each follow-up protocol.

### 2.7. Statistical Analysis

*p* values < 0.05 were considered significant. Differences for continuous variables were analysed with the independent-sample t test for normally distributed variables and with the Mann-Whitney test in case of non-normally distributed variables. Qualitative variables were compared by using chi-squared test or Fisher exact test when applicable.

Because the primary outcome was to evaluate the utility of RL > 2SDS, periodically measured across infancy, childhood and adolescence, in selecting patients at lower risk of developing KI in adulthood (>18 years of age), and not in any age of childhood and adolescence, we decided to use logistic regression instead of Cox regression to calculate the odds ratio (OR) for KI in adulthood of RL > 2SDS at each RUS.

KI-free survival was determined by the Kaplan-Meier method. The day of birth was considered as starting point, while the end point was the date of the KI onset. The patients arriving at their last available follow-up without showing KI were right censored. Kaplan-Meier curves were compared by log-rank test.

We calculated sensitivity, specificity, accuracy, positive and negative likelihood ratio, and positive and negative predictive value (PPV and NPV) of RL > 2SDS at each RUS. Version 26 of SPSS (IBM, Armonk, NY, USA) and version 19.3.1 of MedCalc software (MedCalc Software, Ostend, Belgium) for Windows were used for the statistical analyses.

## 3. Results

### 3.1. Study Population

We included in the study 56 patients (20 females) with age at last follow-up ranging from 18 to 33 years (mean = 21.1 ± 3.4 years) who had received their first evaluation between 1 and 3 months of age. KI was found in 15 patients (26.8%) aged at KI onset between 18 and 26 years (mean age at KI onset = 20.9 ± 3.4 years). Thirteen out of 56 patients (23.2%) had CAKUT of CSFK. Among these 13 patients, five showed non-dilated VUR, three dilated VUR, one UPJO (anterior posterior diameter of the pelvis = 30 mm), and four megaureter (maximal distal ureteral diameter ranging between 7 and 15 mm). CAKUT of CSFK were found in 5/41 patients (12.2%) without, and in 8/15 (53.3%) patients with KI (*p* = 0.001). Three patients showed isolated proteinuria, nine isolated eGFR reduction, one eGFR reduction, proteinuria and hypertension, and two eGFR reduction and hypertension. Among the patients with eGFR reduction, the range of eGFR was 68–88 mL/min/1.73 m^2^, while among the patients with proteinuria the range of urinary protein/creatinine ratio and of albuminuria/creatininuria ratio were 0.3–0.39 mg/mg and 54.2–247.0 µg/mg, respectively. Hypertension was well controlled with ramipril.

The prevalence of RL > 2SDS increased constantly with age, reaching 85.7% in adulthood at a mean age of 21.1 ± 3.4 years (Figure 3A).

The clinical characteristics of patients with and without RL > 2SDS at different ages from birth to adulthood are shown in Table 1. The prevalence of both CAKUT of CSFK (0% vs. 32.5%; *p* = 0.01) and KI (0% vs. 37.5%; *p* = 0.003) was significantly lower in patients with (*n* = 16) compared to those without (*n* = 40) RL > 2SDS at 1–3 months of life. Moreover, patients with RL > 2SDS compared with those with RL < 2SDS at 1–3 months of life presented higher eGFR levels (118.7 ± 9.6 vs. 108.6 ± 14.3; *p* = 0.01) (Table 1).

These findings were confirmed at 1 year of age, while at 2 years a RL > 2SDS was associated only with lower prevalence of CAKUT of CSFK (Table 2). Later in life, a RL > 2SDS was no longer associated with lower prevalence of CAKUT of CSKF, lower prevalence of KI, and higher eGFR levels (Table 2).

### 3.2. Kaplan-Meier Analysis

Patients with RL > 2SDS at 1–3 months of life presented higher cumulative KI-free rate than those with RL < 2SDS (100% vs. 24.8%; *p* = 0.009) (Figure 4A). A higher cumulative KI-free rate was also found in patients with, than in those without, RL > 2SDS at 1 year of age (62.2% vs. 29.6%; *p* = 0.008) (Figure 4B). On the other hand, no significant differences in the Kaplan-Meier curves were shown at the following ages (data shown only for RL at 2 years, Figure 4C, and in adulthood, Figure 4D).

### 3.3. Diagnostic Performance of RL > 2SDS at RUS toward KI in Adulthood (Primary Outcome)

None of the patients with RL > 2SDS at 1–3 months of life presented KI in adulthood. A RL > 2SDS at 1–3 months (OR = infinity) and at 1 year of age (OR = 0.13; 95%CI: 0.03–0.66; *p* = 0.01) was associated with absence of KI in adulthood (Figure 3B). A RL > 2SDS between 2 and 17 years of age was not associated with absence of KI in adulthood (Figure 3B). A RL > 2SDS both at 1–3 months and 1 year of life showed the best specificity (100% and 86.7%, respectively), positive likelihood ratio (infinity and 4.0, respectively) and PPV (100% and 91.7%, respectively) for absence of KI in adulthood (Table 3). With increasing age at RUS, the diagnostic performance of RL > 2SDS reduced, with the exception of sensitivity which improved, reaching 80.5%, 85.4%, and 87.8%, respectively, at 15 years, 17 years, and in adulthood (Table 3).

### 3.4. Cost Analysis (Secondary Outcome)

The economic costs of the different follow-up protocols applied to CSFK children are shown in the Table 4 and ranged from €117,874 with our old approach to €55,860 with the Jawa et al. approach [10]. Among all the protocols, the highest direct costs were related to RUS (costs range: €55,104–€22,960). The indirect costs also negatively impacted the source expenditures, with costs ranging among protocols from €54,818 for the approach of Groen in ‘t Woud et al. [2] to €21,500 for the approach of Jawa et al. [10].

The ERUSA provided a total cost-sparing ranging from 38.6 to 55.3% among the analysed follow-up approaches. The highest cost sparing was obtained in the RUS with an economic save ranging from 85.7 to 65.7%. With ERUSA no patient developing KI in adulthood was missed.

## 4. Discussion

In the present study we showed that, with increase in age, the percentage of CSFK patients reaching RL > 2SDS constantly increases, to up to 85.7% in adulthood, and that when a CSFK reaches a RL > 2SDS it maintains a RL > 2SDS until adulthood. However, CSFK patients presenting RL > 2SDS at birth are certainly different from those reaching RL > 2SDS later in life. The latter, in turn, are different from CSFKs never reaching RL > 2SDS. The difference among these three groups could be the nephronic endowment, which is highest, and fully attained prenatally, for patients born with RL > 2SDS, intermediate for patients reaching RL > 2SDS later in life, and lowest for patients never reaching RL > 2SDS [4]. This could have an impact on the risk of hyperfiltration and KI which is highest for patients never reaching RL > 2SDS and lowest for patients born with RL > 2SDS.

The timing of RUS is critical to select patients with a lower risk of developing KI by the identification of patients with renal hyperplasia.

The longer the time between birth and RUS, higher is probability that a CSFK had been undergone to compensatory hypertrophy. This could determine difficulties in the identification of patients with renal hyperplasia and could reduce the prognostic value of RUS-detected RL. In fact, none of the patients with a RL > 2SDS at 1–3 months of life presented KI after a mean follow-up period of 21.1 years, and a RL > 2SDS was protective against KI into adulthood only until 1 year of age. Later in life, with the increase in the percentage of CSFK with RL > 2SDS, the ability of RUS to select patients at lower risk of KI reduced.

Accordingly, RL > 2SDS at both 1–3 months and 1 year of life had the best diagnostic performance for absence of KI in adulthood with the exception of sensitivity, which improved when increasing the age at RUS.

Moreover, as previously demonstrated in cohorts of CSFK children [3,4], we also confirmed, in a population of patients followed-up until adulthood, an association between early-in-life RL > 2SDS and lower prevalence of CAKUT of CSFK and higher eGFR levels.

With the new recommendations for the management of CSFK, a less aggressive diagnostic approach will be adopted [2]. Therefore, we expect that the utilization of cystography or cysto-scintigraphy and renal scintigraphy will dramatically reduce, with subsequent diagnosis only of the most severe and symptomatic CAKUT of CSFK [23]. For this reason, we expect that most of the decision-making processes for follow-up timing of CSFK will be based only on the RUS among instrumental exams. From this viewpoint, our findings could improve the RUS prognostic performance, taking advantage of the information provided by the RL of CSFK in the first months of life. In order to test the diagnostic performance of RUS in mimicking this future scenario of the daily clinical practice, we decided to not run a multivariate logistic regression including the other main risk factors of KI, such as CAKUT of CSFK.

Recently, Jawa et al. showed that he development of a specific algorithm could reduce by 47% the proportion of avoidable ultrasounds ordered in CSFK children resulting in about $46,000 of annual savings [10]. Our cost analysis showed that ERUSA could further reduce the costs when applied to the analyzed follow-up protocols without missing long-term important prognostic information (Table 4).

We included in this algorithm only the RUS giving useful prognostic information (at 1–3 months and 1 year of age). In addition to RUS at 1–3 years of age, ERUSA established RUS at 1 year of age, because at this age we still observed significant prognostic information, which could further orientate clinicians towards future prognosis. However, because two patients with RL > 2SDS at 1 year of age developed KI in adulthood, we suggested following-up the patients independently from the RL at this age. As the RUS after 1 year of age did not give useful prognostic information, ERUSA established a process of no longer submitting patients to RUS after this age in case of absence of complex features.

ERUSA could be of interest for all specialists managing CSFK patients because it is potentially applicable to any follow-up protocol of CSFK, with significant impact on cost saving.

The increase in prenatal diagnoses determines “prenatal” and “postnatal” anxiety in parents of CSFK patients [24,25,26]. ERUSA could precociously identify CSFK patients with the best prognosis, with reassurance for parents in the first months of life and afterwards.

It was noteworthy that our patients attained adulthood showing both lower prevalence and severity of KI than previously reported [27,28].

The poorer CSFK outcomes described in the literature could in part ensue from the oversampling of the most severely affected patients due to the enrolment of subjects observed at any age just because they showed signs of an already established renal injury [27,28].

In contrast, in the present study we enrolled only patients with prenatal diagnosis and very early evaluation before the onset of any sign of KI. Our data, however, strengthen the usefulness of early RUS as predictors of later prognosis of CSFK, even when facing patients with low prevalence and severity of KI.

As a future study perspective, it could be interesting to enrol larger cohorts of patients with renal agenesis and multi-cystic dysplastic kidney in order to evaluate if differences in the prognostic utility of early-in-life RUS could exist.

In fact, recently, Matsell et al. demonstrated that the outcomes of renal agenesis are different than multi-cystic dysplastic kidney, probably due to the different pathogenesis, which appears to be linked to a failure of ureteric duct development for renal agenesis and to an interruption in established kidney induction after it has already been initiated for the multi-cystic dysplastic kidney [29].

Strengths of this study are represented by the first, careful follow-up from birth to adulthood and by the evaluation of RL > 2SDS in different moments of infancy, childhood and adolescence as predictor of absence of KI in adulthood. Moreover, this is the first study extensively analysing the follow-up costs of CSFK patients.

The main limitation of the study is represented by the limited sample size. Due to this, the study could be underpowered in evaluating the predictivity of RUS, especially from 2 years of age.

Another limitation is represented by the vast change concerning the quality and accuracy of ultrasound imaging during recent decades. All the enrolled patients, however, underwent extensive nephro-urological evaluation at enrolment (1–3 months of life) by execution, in addition to RUS, of Tc99mMAG3 or Tc99mDMSA scintigraphy and voiding cystourethrogram (in males) or cysto-scintigraphy (in females). This extensive initial evaluation of the patients, in our opinion, could have limited the associated CAKUT misdiagnoses, possibly related to the limited quality and accuracy of ultrasound imaging of the older renal ultrasounds.

Moreover, our study mainly focalizes on the RL > 2SDS measured at RUS, as predictor of absence of KI and the RL assessment by RUS has shown high reproducibility compared with other measurements in different studies over recent decades [30,31,32,33].

## 5. Conclusions

In conclusion, a RL > 2SDS at 1–3 months of life identifies the population of CSFK patients with the best prognosis, and at 1 year of age it selects patients with reduced risk of KI. Later in life, the prognostic information available from RUS is limited. Follow-up assessment for KI in subjects with RL > 2SDS at 1–3 months of life and absence of complex features could not be performed until adulthood. Following further validation in multicenter studies with higher sample size, ERUSA could be implemented in daily clinical practice to reduce costs while maintaining long-term prognostic information.

## Figures and Tables

**Figure 1 jcm-11-01052-f001:**
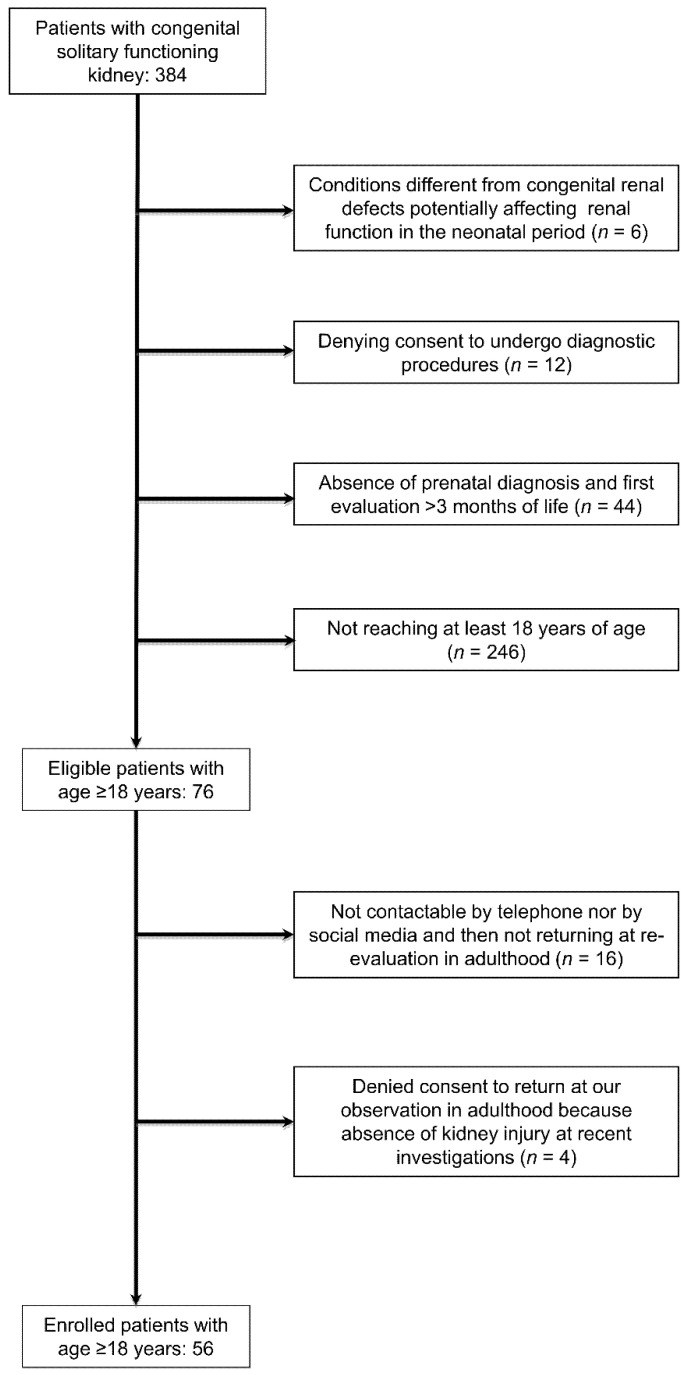
Consort diagram describing patients’ enrolment.

**Figure 2 jcm-11-01052-f002:**
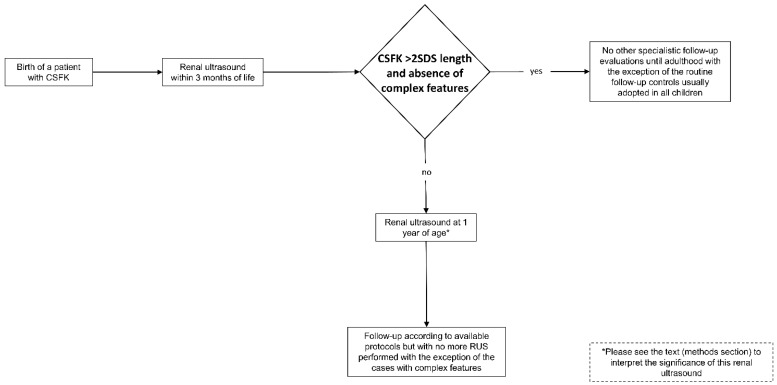
Early renal ultrasound algorithm (ERUSA).

**Figure 3 jcm-11-01052-f003:**
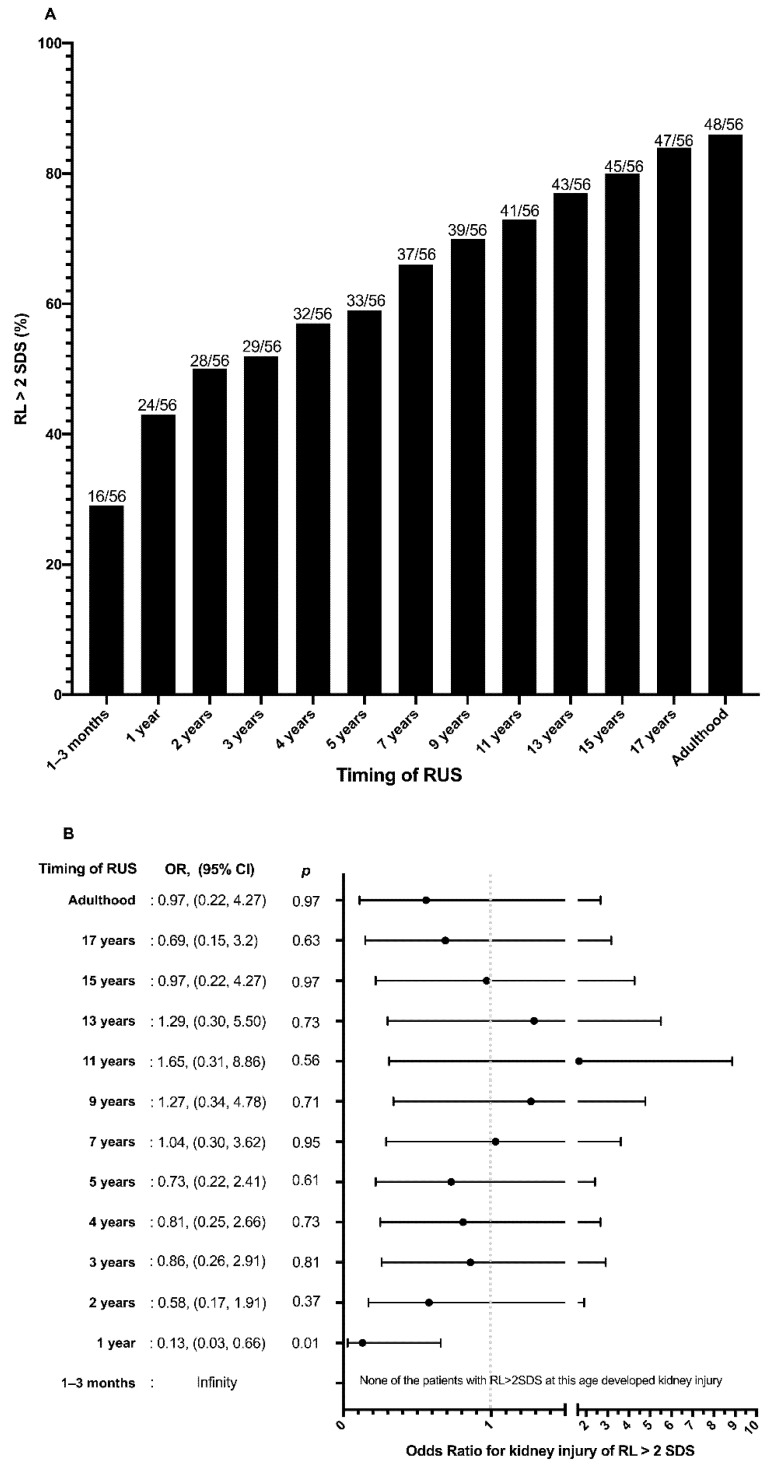
(**A**) Percentage of patients with congenital solitary functioning kidney (CSFK) reaching renal length (RL) > 2 standard deviations (SDS) at periodical renal ultrasound performed from birth to adulthood. (**B**) Odds ratio for kidney injury (KI) of renal length (RL) > 2 standard deviations (SDS) at periodical renal ultrasound performed from birth to adulthood.

**Figure 4 jcm-11-01052-f004:**
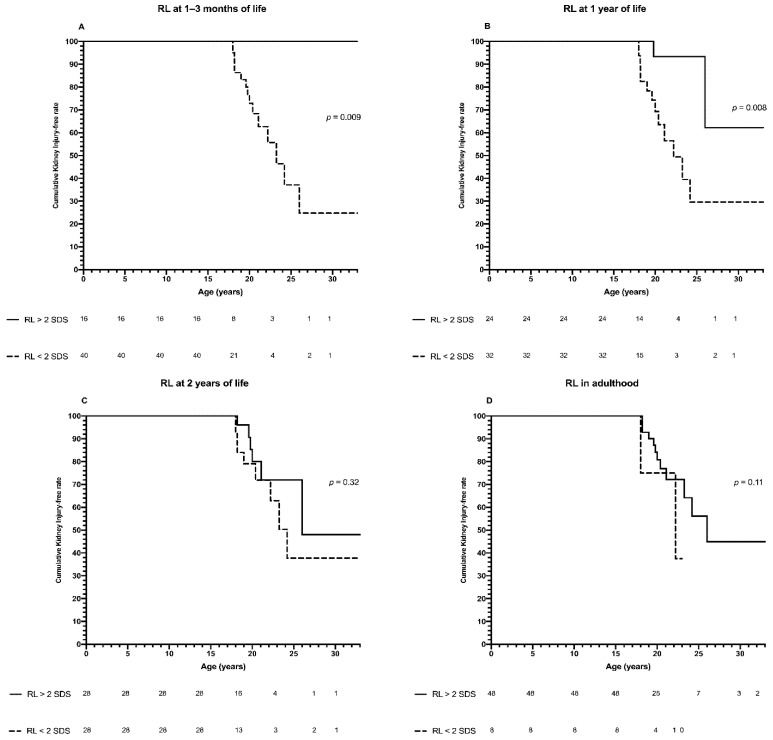
Kaplan-Meier analysis. Panel (**A**): Cumulative kidney injury-free rate of patients with congenital solitary functioning kidney (CSFK) on the basis of renal length (RL) > or <2 standard deviations (SDS) at 1–3 months of life. The cumulative kidney injury-free rate was 100% until 33 years of age for patients with RL > 2SDS, while for patients with RL < 2SDS it was 100% at 15 years, 72.9% at 20 years, 37.1% at 25 years, 24.8% at 33 years of age (*p* = 0.009). Panel (**B**): Cumulative kidney injury-free rate of patients with congenital solitary functioning kidney (CSFK) on the basis of renal length (RL) > or <2 standard deviations (SDS) at 1 year of life. The cumulative kidney injury-free rate was 100% until 18 years, 93.3% from 20 to 25 years and 62.2% at 30 and 33 years of age for patients with RL > 2SDS while for patients with RL < 2SDS it was 100% at 15 years, 69.3% at 20 years, 29.6% from 25 years to 33 years of age (*p* = 0.008). Panel (**C**): Cumulative kidney injury-free rate of patients with congenital solitary functioning kidney (CSFK) on the basis of renal length (RL) > or <2 standard deviations (SDS) at 2 years of life. The cumulative kidney injury-free rate was 100% until 18 years, 80% at 20 years, 72% at 25 years and 48% at 33 years of age for patients with RL > 2SDS while for patients with RL < 2SDS it was 100% at 15 years, 92.8% at 18 years, 79.1% at 20 years, and 37.7% from 25 to 33 years of age (*p* = 0.32). Panel (**D**): Cumulative kidney injury-free rate of patients with congenital solitary functioning kidney (CSFK) on the basis of renal length (RL) > or <2 standard deviations (SDS) in adulthood. The cumulative kidney injury-free rate was 100% until 18 years, 80.9% at 20 years, 56.2% at 25 years, and 44.9% at 33 years, while for patients with RL < 2SDS it was 100% at 15 years, 75% from 18 to 20 years and 37.5% afterwards (*p* = 0.11).

**Table 1 jcm-11-01052-t001:** Characteristics of the population with and without RL > 2SDS at 1–3 months of life.

	RL < 2SDS (*n* = 40)	RL > 2SDS (*n* = 16)	*p*
Female gender, No. (%)	16 (40)	4 (25)	0.49
MCDK, No. (%)	23 (57.5)	7 (43.8)	0.35
Birth weight, mean (SDS), g	3100 (612)	3180 (490)	0.64
Birth weight < 2.500 kg, %	6 (15.0)	2 (12.5)	0.80
CAKUT of CSFK, No. (%)	13 (32.5)	0	0.01
Febrile urinary tract infections, No. (%)	3 (7.5)	0	0.26
Age at last follow-up, median (CI), year	20.0 (19.0/21.8)	19.9 (18.4/23.0)	0.68
SUA, mean (SDS), mg/dL	5.2 (1.4)	5.5 (1.5)	0.51
SBP, mean (SDS), mmHg	116.9 (11.5)	119.1 (11.4)	0.52
DBP, mean (SDS), mmHg	66.0 (10.1)	69.5 (7.2)	0.27
SBP_24 h_, median (CI), mmHg	114.2 (111.0/124.3)	111.3 (105.1/122.4)	0.78
DBP_24 h_, mean (SDS), mmHg	69.0 (6.0)	67.1 (5.7)	0.41
SBP_day_, mean (SDS), mmHg	119.0 (10.6)	115.6 (10.0)	0.24
DBP_day_, mean (SDS), mmHg	70.8 (6.1)	69.3 (6.8)	0.42
SBP_night_, mean (SDS), mmHg	108.8 (11.7)	104.8 (9.8)	0.25
DBP_night_, mean (SDS), mmHg	60.7 (7.1)	61.7 (5.2)	0.63
Systolic dipping, median (CI), %	8.4 (5.0/12.0)	8.3 (4.5/12.5)	0.98
Diastolic dipping, mean (SDS), %	13.5 (6.8)	9.6 (8.3)	0.08
Systolic load, median (CI), %	11.9 (6.9/24.0)	8.0 (0/14.9)	0.12
Diastolic load, median (CI), %	20.0 (9.5/27.0)	16.6 (6.8/25.0)	0.95
eGFR, mean (SDS), mL/min/1.73 m^2^	108.6 (14.3)	118.7 (9.6)	0.01
UPr/Cr, median (CI), mg/mg	0.06 (0.05/0.09)	0.06 (0.05/0.10)	0.63
Albuminuria, median (CI), mg/L	15.5 (10.0/14.9)	14.0 (9.0/23.5)	0.95
Ua/Cr, median (CI), µg/mg	10.7 (6.3/14.9)	6.4 (5.7/15.2)	0.30
Kidney Injury, No. (%)	15 (37.5)	0	0.003
Renal compensatory hypertrophy at last follow up, No. (%)	32 (80)	16 (100)	0.09

Continuous variables are presented as median and interquartile range if not normally distributed and as mean and SDS if normally distributed. Abbreviations: CAKUT, congenital anomalies of the kidneys and urinary tract; CSFK, congenital solitary functioning kidney; DBP, diastolic blood pressure; eGFR, estimated glomerular filtration rate; MCDK, multi-cystic dysplastic kidney; SBP, systolic blood pressure; SDS, standard deviation score; SUA, serum uric acid; Ua/Cr, urinary albumin/creatinine ratio; UPr/Cr, urinary protein/creatinine ratio.

**Table 2 jcm-11-01052-t002:** Characteristics of the population with and without RL > 2SDS at different ages from birth to adulthood.

	**1–3 Months**			**1 Year**			**2 Years**		
	**RL < 2SDS** **(*n* = 40)**	**RL > 2SDS** **(*n* = 16)**	** *p* **	**RL < 2SDS** **(*n* = 32)**	**RL > 2SDS** **(*n* = 24)**	** *p* **	**RL < 2SDS** **(*n* = 28)**	**RL > 2SDS** **(*n* = 28)**	** *p* **
CAKUT of CSFK, No. (%)	13 (32.5)	0	0.01	11 (34.4)	2 (8.3)	0.02	10 (35.7)	3 (10.7)	0.03
eGFR, mean (SDS), mL/min/1.73 m^2^	108.6 (14.3)	118.7 (9.6)	0.01	107.3 (14.7)	116.8 (10.3)	0.01	108.6 (14.8)	114.3 (12.4)	0.12
Kidney Injury, No. (%)	15 (37.5)	0	0.003	13 (40.6)	2 (8.3)	0.007	9 (32.1)	6 (21.4)	0.36
	**3 Years**			**4 Years**			**5 Years**		
	**RL < 2SDS** **(*n* = 27)**	**RL > 2SDS** **(*n* = 29)**	** *p* **	**RL < 2SDS** **(*n* = 24)**	**RL > 2SDS** **(*n* = 32)**	** *p* **	**RL < 2SDS** **(*n* = 23)**	**RL > 2SDS** **(*n* = 33)**	** *p* **
CAKUT of CSFK, No. (%)	9 (33.3)	4 (13.8)	0.09	8 (33.3)	5 (15.6)	0.12	8 (34.8)	5 (15.1)	0.09
eGFR, mean (SDS), mL/min/1.73 m^2^	108.6 (15.1)	114.1 (12.2)	0.14	108.8 (14.9)	112.7 (13.0)	0.42	109.1 (14.8)	113.1 (13.0)	0.29
Kidney Injury, No. (%)	8 (29.6)	7 (24.1)	0.64	7 (29.2)	8 (25.0)	0.72	7 (30.4)	8 (24.2)	0.61
	**7 Years**			**9 Years**			**11 Years**		
	**RL < 2SDS** **(*n* = 19)**	**RL > 2SDS** **(*n* = 37)**	** *p* **	**RL < 2SDS** **(*n* = 17)**	**RL > 2SDS** **(*n* = 39)**	** *p* **	**RL < 2SDS** **(*n* = 15)**	**RL > 2SDS** **(*n* = 41)**	** *p* **
CAKUT of CSFK, No. (%)	6 (31.6)	7 (18.9)	0.28	6 (64.7)	7 (17.9)	0.16	5 (33.3)	8 (19.5)	0.28
eGFR, mean (SDS), mL/min/1.73 m^2^	109.7 (15.8)	112.4 (12.8)	0.27	111.3 (14.7)	111.5 (13.6)	0.96	111.3 (15.7)	111.5 (13.3)	0.95
Kidney Injury, No. (%)	5 (26.3)	10 (27.0)	0.95	4 (23.5)	11 (28.2)	0.72	4 (26.7)	11 (26.8)	0.99
				**Adulthood**					
				**RL < 2SDS** **(*n* = 8)**		**RL > 2SDS** **(*n* = 48)**		** *p* **	
CAKUT of CSFK, No. (%)				2 (25.0)		11 (22.9)		0.90	
eGFR, mean (SDS), mL/min/1.73 m^2^				114.0 (18.4)		111.1 (13.1)		0.58	
Kidney Injury, No. (%)				3 (37.5)		12 (25.0)		0.46	

Continuous variables are presented as median and interquartile range if not normally distributed and as mean and SDS if normally distributed. Abbreviations: CAKUT, congenital anomalies of the kidneys and urinary tract; CSFK, congenital solitary functioning kidney; eGFR, estimated glomerular filtration rate.

**Table 3 jcm-11-01052-t003:** Prognostic accuracy of RL > 2SDS for absence of kidney injury in adulthood evaluated in different pediatric ages.

RL > 2SDS at Different Ages	True Positive: False Positive	True Negative: False Negative	Sensitivity (95% CI)	Specificity (95% CI)	Accuracy (95% CI)	Positive Likelihood Ratio (95% CI)	Negative Likelihood Ratio (95% CI)	Positive Predictive Value (95% CI)	Negative Predictive Value (95% CI)
1–3 months	16:0	15:25	39% (24.2–55.5)	100% (78.2–100)	55.4% (41.5–68.7)	Infinity	0.6 (0.5–0.8)	100%	37.5% (31.9–43.4)
1 year	22:2	13:19	53.7% (37.4–69.3)	86.7 (59.5–98.3)	62.5% (48.5–75.1)	4.0 (1.1–15.1)	0.5 (0.4–0.8)	91.7% (74.6–97.6)	40.6% (31.8–50.1)
2 years	22:6	9:19	53.7% (37.4–69.3)	60.0% (32.3–83.7)	55.4% (41.5–68.7)	1.3 (0.7–2.6)	0.8 (0.5–1.3)	78.6% (65.0–87.9)	32.1% (21.8–44.5)
3 years	22:7	8:19	53.7% (37.4–69.3)	55.3% (26.6–78.7)	53.6% (39.7–67.0)	1.1 (0.6–2.1)	0.9 (0.5–1.5)	75.9% (63.0–85.3)	29.6% (19.1–42.8)
4 years	24:8	7:17	58.5% (42.1–73.7)	46.7% (21.3–73.4)	55.4% (41.5–68.7)	1.1 (0.6–1.9)	0.9 (0.5–1.7)	75% (63.6–83.7)	29.2 (17.7–44.1)
5 years	25:8	7:16	61.0% (44.5–75.8)	46.7% (21.3–73.4)	57.1% (43.2–70.3)	1.1 (0.7–2.0)	0.8 (0.4–1.6)	75.8% (64.7–84.2)	30.4% (18.4–45.9)
7 years	27:10	5:14	65.8% (49.4–79.9)	33.3% (11.8–61.6)	57.1% (43.2–70.3)	1.0 (0.6–1.5)	1.0 (0.4–2.3)	73.0% (63.9–80.4)	26.3% (13.4–45.1)
9 years	28:11	4:13	68.3% (51.9–81.9)	26.7% (7.8–55.1)	57.1% (43.2–70.3)	0.9 (0.6–1.3)	1.2 (0.5–3.1)	71.8% (63.8–78.6)	23.5% (10.6–44.4)
11 years	30:11	4:11	73.2% (57.1–85.8)	26.7% (7.8–55.1)	60.7% (46.7–73.5)	1 (0.7–1.4)	1 (0.4–2.7)	73.2 (65.6–79.6)	26.7 (12.0–49.2)
13 years	31:12	3:10	75.6% (59.7–87.6)	20.0% (4.3–48.1)	60.7% (46.7–73.5)	0.9 (0.7–1.3)	1.2 (0.4–3.8)	72.1% (65.5–77.8)	23.1 (8.7–48.6)
15 years	33:12	3:8	80.5% (65.1–91.2)	20.0% (4.3–48.1)	64.3% (50.4–76.6)	1.0 (0.7–1.3)	0.9 (0.3–3.2)	73.3% (67.2–78.7)	27.3% (10.3–55.2)
17 years	35:12	3:6	85.4% (70.8–94.4)	20.0% (4.3–48.1)	67.9% (54.0–79.7)	1.1 (0.8–1.4)	0.7 (0.2–2.6)	74.5% (68.7–79.5)	33.3% (12.5–63.6)
Adulthood	36:12	3:5	87.8% (73.8–95.9)	20.0% (4.3–48.1)	69.6% (55.9–81.2)	1.1 (0.8–1.4)	0.6 (0.2–2.2)	75.0% (69.4–79.8)	37.5 (14.0–68.8)

**Table 4 jcm-11-01052-t004:** Impact of the ERUSA on economic costs of different follow-up approaches that can be adopted to order follow-up evaluations in children with CSFK.

	Old Approach Adopted in Our Clinic	Old Approach Modified on the Basis of ERUSA	Groen in ‘t Woud et al. Approach [2]	Groen in ‘t Woud et al. Approach Modified on the Basis of ERUSA	Jawa et al. Approach [10]	Jawa et al. Approach Modified on the Basis of ERUSA
RUS, number	672	96	280	96	336	96
RUS costs, €	55,104	7872	22,960	7872	27,552	7872
RUS cost saving, %	–	85.7	–	65.7	–	71.4
Urinalysis, number	672	480	1008	720	280	200
Urinalysis costs, €	1458	1041	2187	1562	607	434
Urinalysis cost saving, %	–	28.6	–	28.6	–	28.5
Creatinine dosage, number	672	480	376	280	112	80
Creatinine dosage costs, €	2493	1781	1395	1039	416	297
Creatinine dosage cost saving, %	–	28.6	–	25.6	–	28.6
Follow-up visits with blood pressure measurement, number	672	480	1008	720	336	200
Follow-up visits costs, €	13,883	9916	20,825	14,875	5785	4132
Follow-up visits cost saving, %	–	28.6	–	28.6	–	28.6
Indirect costs, €	44,936	32,097	54,818	37,357	21,500	13,097
Indirect costs saving, %	–	28.6	–	31.8	–	39.1
Total costs, €	117,874	52,707	102,185	62,705	55,860	25,832
Total costs saving, %	–	55.3	–	38.6	–	53.8
Number of patients with missed KI	0	0	0	0	0	0

## Data Availability

The datasets generated during and/or analysed during the current study are available from the corresponding author on reasonable request.

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
