# Peer review of "Early Renal Ultrasound in Patients with Congenital Solitary Kidney Can Guide Follow-Up Strategy Reducing Costs While Keeping Long-Term Prognostic Information"

_jcm, 2022, doi:10.3390/jcm11041052_

Round 1

Reviewer 1 Report

The authors report in a structured and clear manner their findings in patients with congenital solitary functioning kidney (CSFK). They manage to proove the assumption, that a renal length (RL)>2 standard deviation scores in the first three months of live, predicts a favorable outcome without the risk of kidney injury. 

They were able to analyse data of 56 adult patients with CSFK including renal length measurements at different defined time points. The statistic power of their findings is only hampered by the overall low numbers of patients.

Before deciding not to monitor these patients based on the reported analysis, a multicenter study with a correspondingly higher number of cases would be desirable. That should be adressed in the discussion/limitations.

Moreover, the reader might be interested in the details of CAKUT and if there was any correlation between severeness of CAKUT and KI later on.

Author Response

Reviewer 1

The authors report in a structured and clear manner their findings in patients with congenital solitary functioning kidney (CSFK). They manage to prove the assumption, that a renal length (RL)>2 standard deviation scores in the first three months of live, predicts a favorable outcome without the risk of kidney injury. 

They were able to analyze data of 56 adult patients with CSFK including renal length measurements at different defined time points. The statistic power of their findings is only hampered by the overall low numbers of patients. Before deciding not to monitor these patients based on the reported analysis, a multicenter study with a correspondingly higher number of cases would be desirable. That should be addressed in the discussion/limitations.

Answer: we agree with you, we have acknowledged this limitation among limitations of the study (please see lines 397-399 of the new version of the manuscript). We also specified in the conclusion section that ERUSA could be implemented in the daily clinical practice after further validation in multicenter studies with higher sample size (please see lines 416-417 of the new version of the manuscript).

Moreover, the reader might be interested in the details of CAKUT and if there was any correlation between severeness of CAKUT and KI later on.

Answer: because of the very limited number of patients with CAKUT after classification in subcategories the correlation between these categories and kidney injury could be extremely underpowered. Therefore, to avoid misleading information to the Readers we decided to not add these analyses in the new version of the manuscript. However, following your comment we added more details about the found CAKUT (please see lines 217-220 of the new version of the manuscript).

Reviewer 2 Report

Dear authors, 
Your work introduces the notion that early renal ultrasound provides prognostic information concerning renal function at a later age. This is further associated with cost-effectiveness. 

I have the following comments: 
Major: 
L99-114: It would be helpful for the readers to present (as supplementary material) which anatomic anomalies are present, given the rarity of the disorder. 
L 80-84: Evaluation of kidney function including the performance of renal ultrasound was performed in a mean time frame of 21,1 years. During that time there has been a vast change concerning the quality and accuracy of ultrasound imaging. How did the authors correct for the less qualitative imaging during the last decade of the 20th century compared with future imaging? I am of the opinion that this introduces a confounding factor that may possibly influence outcome interpretation.
L129-136: The definition of kidney injury is confusing.  Due to the definition provided by the authors, KI could be present only by the presence of arterial hypertension or isolated proteinuria without eGFR change. The notion is further complicated by the fact that hydronephrosis is sometimes associated with low-grade proteinuria originating from hydronephros. Another point is the fact that kidney injury presents a continuum. One refers to it as acute kidney injury a week with renal function deterioration, acute kidney disease till 3 months, and after 3 months as chronic kidney disease (Chawla et al. Nat. Rev. Nephrol. 2017, vol. 13 pg 241-257). Due to this opinion the majority of your patients had Stage I chronic kidney disease regardless of renal dimensions.  And hence your patients present the same risk of renal function deterioration. How can you justify that renal ultrasound provides prognostic information?

When you only include patients presenting eGFR reduction while ignoring the patients presenting isolated proteinuria which are your results?

Statistical Analysis: Why wasn´t Cox Regression Analysis used?

Minor: 
L181: Please refer the SPSS and Med Clac Software version. 

Please provide non-blurred images. 

All of my best regards

Author Response

Reviewer 2

Dear authors, 
Your work introduces the notion that early renal ultrasound provides prognostic information concerning renal function at a later age. This is further associated with cost-effectiveness. 

I have the following comments: 
Major: 
L99-114: It would be helpful for the readers to present (as supplementary material) which anatomic anomalies are present, given the rarity of the disorder. 

Answer: Thirteen out of 56 patients (23.2%) had CAKUT of CSFK. Among these 13 patients, 5 patients showed non-dilated VUR, 3 patients dilated VUR, 1 patient UPJO (anterior posterior diameter of the pelvis=30 mm), and 4 patients megaureter (maximal distal ureteral diameter ranging between 7 and 15 mm). We added this information in the new version of the manuscript, please see lines 217-220. Because JCM has no limits in manuscript length and because this information can be reassumed in few lines, we added this information in the results section instead of in supplementary material.

L 80-84: Evaluation of kidney function including the performance of renal ultrasound was performed in a mean time frame of 21,1 years. During that time there has been a vast change concerning the quality and accuracy of ultrasound imaging. How did the authors correct for the less qualitative imaging during the last decade of the 20th century compared with future imaging? I am of the opinion that this introduces a confounding factor that may possibly influence outcome interpretation. 

Answer: All the enrolled patients underwent extensive nephro-urological evaluation at enrolment (1–3 months of life) by execution, in addition to renal ultrasound, of Tc99mMag3 or Tc99mDMSA scintigraphy and voiding cystourethrogram (in males) or cystoscintigraphy (in females). This extensive initial evaluation of the patients, in our opinion, has limited CAKUT misdiagnoses possibly related to the limited quality and accuracy of ultrasound imaging of the older renal ultrasounds. Moreover, our study mainly focalizes on the RL>2SDS measured at RUS as predictor of absence of KI and the RL assessment by RUS has shown high reproducibility compared with other measurements in different studies in the last decades (1.Braconnier, P.; Piskunowicz, M.; Vakilzadeh, N.; Müller, M.E.; Zürcher, E.; Burnier, M.; Pruijm, M. How reliable is renal ultrasound to measure renal length and volume in patients with chronic kidney disease compared with magnetic resonance imaging? Acta radiol. 2020, 61, 117–127, doi:10.1177/0284185119847680. 2.Ablett, M.J.; Coulthard, A.; Lee, R.E.J.; Richardson, D.L.; Bellas, T.; Owen, J.P.; Keir, M.J.; Butler, T.J. How reliable are ultrasound measurements of renal length in adults? Br. J. Radiol. 1995, 68, 1087–1089, doi:10.1259/0007-1285-68-814-1087; 3.Bakker, J.; Olree, M.; Kaatee, R.; De Lange, E.E.; Moons, K.G.M.; Beutler, J.J.; Beek, F.J.A. Renal Volume Measurements: Accuracy and Repeatability of US Compared with That of MR Imaging1. https://doi.org/10.1148/radiology.211.3.r99jn19623 1999, 211, 623–628, doi:10.1148/RADIOLOGY.211.3.R99JN19623.; Emamian, S.A.; Nielsen, M.B.; Pedersen, J.F.; Ytte, L. Kidney dimensions at sonography: correlation with age, sex, and habitus in 665 adult volunteers. AJR. Am. J. Roentgenol. 1993, 160, 83–86, doi:10.2214/AJR.160.1.8416654.

We added these considerations among limitations of the study (please see lines 400–410 of the new version of the manuscript).

L129-136: The definition of kidney injury is confusing.  Due to the definition provided by the authors, KI could be present only by the presence of arterial hypertension or isolated proteinuria without eGFR change. The notion is further complicated by the fact that hydronephrosis is sometimes associated with low-grade proteinuria originating from hydronephrosis. Another point is the fact that kidney injury presents a continuum. One refers to it as acute kidney injury a week with renal function deterioration, acute kidney disease till 3 months, and after 3 months as chronic kidney disease (Chawla et al. Nat. Rev. Nephrol. 2017, vol. 13 pg 241-257). Due to this opinion the majority of your patients had Stage I chronic kidney disease regardless of renal dimensions.  And hence your patients present the same risk of renal function deterioration. How can you justify that renal ultrasound provides prognostic information?

Answer: we decided to use the presented definition of kidney injury to be consistent with the most important reports about CSFK such as the KIMONO study and other studies (doi: 10.2215/CJN.07870812; doi: 10.1093/ndt/gfq844; doi: 10.1016/j.juro.2016.03.173; https://doi.org/10.1007/s00467-018-4111-3; doi: 10.1016/j.juro.2017.05.076; http://dx.doi.org/10.1016/j.euros.2021.01.003).

Adopting this definition, we are able to identify patients presenting signs indicating an impaired renal function (declined eGFR, proteinuria or hypertension) as result of the CSFK-related reduced nephronic mass.

According to KDIGO CKD guidelines (Kidney International Supplements, 2013, 3, vii) also only albuminuria more or equal to 30 mg/g persistent for more than 3 months is a criterium to define CKD. Therefore also if –as you rightly stated– hydronephrosis could be complicated by low-grade proteinuria, we believe that if this proteinuria is persistent for 3 months, it respects the KDIGO definition. In all our patients with proteinuria we documented a persistence of proteinuria for at least 3 months. We better specified in the new version of the manuscript that not only for proteinuria but also for eGFR and hypertension we considered the persistence for >3 months before to define the patient as affected by kidney injury (please see lines 146-153 of the new version of the manuscript).

In our population 15 patients showed signs of kidney injury while 41 patients did not show nor declined eGFR nor hypertension nor proteinuria. Only a RL>2SDS at 1-3 months of life was able to select patients not showing any sign kidney injury at last follow-up in adulthood. Similarly, also a RL>2SDS at 1 year of age was able to select patients at lower risk of developing KI in adulthood. Therefore, examining our data, the prognostic information of renal ultrasound is provided only at 1-3months and 1 year of life.

We agree with you that the majority of our patients, such as the majority of patients with CSFK worldwide, could be labelled as affected by stage 1 CKD, because according to KDIGO definition they present a structural abnormality detected by imaging, especially if associated to CAKUT on the solitary functioning kidney. However, not all these patients will develop declined eGFR or albuminuria, therefore, in our opinion, to select patients at higher risk and needing of stricter follow-up could be useful to optimize the resources.

 When you only include patients presenting eGFR reduction while ignoring the patients presenting isolated proteinuria which are your results?

Answer: for the explanations given in the previous answer (to give a definition consistent with the previous international reports) we prefer to not eliminate the proteinuria form the definition of kidney injury. However, analyzing the data ignoring the 3 patients with isolated proteinuria we found similar results with an OR not calculable for patients with RL >2SDS at 1-3 months of life, a significant OR for patients with RL>2SDS at 1 year of age [OR= 0.41 (95%CI 0.11/1.59;p=0.02)], and a no more significant OR for RL >2SDS later in life.

Statistical Analysis: Why wasn´t Cox Regression Analysis used?

Answer: Because the primary outcome was to evaluate the utility of RL>2SDS –periodically measured across infancy, childhood and adolescence– in selecting patients at lower risk of developing KI in adulthood (>18years of age) and not in any age of childhood and adolescence, we decided to use logistic regression instead of Cox regression to calculate the OR for KI in adulthood of RL>2SDS at each RUS. We added this information in the new version of the manuscript (please see lines 198-202 of the new version of the manuscript).

Minor: 
L181: Please refer the SPSS and Med Calc Software version. 

Answer: we added this information in the new version of the manuscript. Please see lines 208-209.

Please provide non-blurred images. 

Answer: we substituted all the images of the new version of the manuscript with high resolution ones.

Reviewer 3 Report

The manuscript "Early renal ultrasound in patients with congenital solitary kidney can guide follow-up strategies reducing costs while keeping long-term prognostic information" is interesting and focus on the relevance of early diagnosis and prognosis of congenital solitary functioning kidney in order to reduce the costs, since the patients will need lifelong urological care.

The manuscript is interesting, however some aspects can be improved_

  • The introduction is confusing, needs to be improved;
  • In materials and methods, the patientes ere selected from which hospitals?
  • In materials and methods, the authors claimed that "The sutudy obtined ethical aprroval" but from whom? The hospital ethical comitte? Please clarify and provide the data
  • The sample size is really small to extrapolate the results.
  • The disccusion is confusing in the discussion of the results.

Author Response

Reviewer 3

The manuscript "Early renal ultrasound in patients with congenital solitary kidney can guide follow-up strategies reducing costs while keeping long-term prognostic information" is interesting and focus on the relevance of early diagnosis and prognosis of congenital solitary functioning kidney in order to reduce the costs, since the patients will need lifelong urological care. 

The manuscript is interesting, however some aspects can be improved.

-The introduction is confusing, needs to be improved

Answer: we modified the introduction section pointing more accurately on the topic of the study. Please see lines 49-62 of the new version of the manuscript.

-In materials and methods, the patients are selected from which hospitals?

Answer: we added this information in the new version of the manuscript, please see lines 82-83.

-In materials and methods, the authors claimed that "The study obtained ethical approval" but from whom? The hospital ethical committee? Please clarify and provide the data

Answer: we added this information in the new version of the manuscript, please see lines 85-87.

-The sample size is really small to extrapolate the results.

Answer: when a sample size is small, the study can be underpowered and significant differences could not be detected. The fact that we obtained statistically significant results underlines that we faced very strong associations between RL>2SDS at 1-3 months and 1 year of age of life and absence of kidney injury in adulthood. However, we are conscious of this limit and we acknowledged it among limitations of the study (please see lines 397-399 of the new version of the manuscript). Moreover, we mitigated the conclusions (please see lines 416-417 of the new version of the manuscript).

-The discussion is confusing in the discussion of the results

 Answer: we edited all the discussion section, please see lines 316-418 of the new version of the manuscript.

Reviewer 4 Report

Thank you very much for your nice study. I consider it of a great merit taking into account the length of patients’ follow-up.

The objectives are relevant and results are clearly exposed. Most of minor changes are suggested rather than mandatory, or just additional explanations. Please consider them, if possible taking into account limits for extension of the article.

  1. In a relatively recent paper published in Pediatric Nephrology, some prognostic differences in CSFK have been suggested between renal agenesia and MCDK. Those patients are commonly grouped for studies on CSFK. This does not invalidate your study as proportion of MCDK in <2 and >2SDS is reflected in Table 1 (and no significant differences are found) and, besides, prognostic factors associated with poorer outcomes in MCDK are related to CAKUT and lack of compensatory hypertrophy, which have been considered. But maybe this fact would merit a comment.
  2. Age at last evaluation in clearly stated. Because these are relatively young adults, I think that utility of ERUSA in differentiating patients with good prognosis should be clearly stated in Summary as “In conclusion, only a RL>2SDS at 1–3months and 1year of age predicted good prognosis in YOUNG adulthood.” Anyway, you prudently recommend follow-up during adulthood, even in patients with early compensatory hypertrophy, which is reasonable.
  3. Do not use the term “urinary microalbumin” as microalbumin does not exist as a substance and KDIGO advices against the term “microalbuminuria”, use “urinary albumin” or “albuminuria”.
  4. If possible, different methods for creatinine measurements should be explained (probably because of changes in lab techniques across the years). Same for the use of original Schwartz formula.
  5. Definitions for different CAKUT diagnosis are arguable but acceptable. Specifically talking about UPJO, I would rather use the term “compatible” rather than “positive” MAG3 scintigraphy, because it is a bit confusing. I assume you are talking about type 2 curves with a plateau in excretion, I do not know whether your urologist consider for surgery without derangement in relative kidney function. Type MAG3, not Mag3.
  6. Expertise of radiologists and specialists in Nuclear Medicine in studies on pediatric patients should be stated, as well as how many of them perform this test (length measurements can vary from one observer to other). This would give readers an idea about the reproducibility on their own facilities, as accuracy could change with studies interpreted by non-expert specialists.
  7. Time spent on procedures could be arguable (maybe overestimated in my personal opinion for lab procedures as determinations are performed at the same time for multiple samples) but I would accept it as it is well explained.
  8. Please type “Windows” (the operating system) with initial capital letter.
  9. Post-hoc power calculations can be tricky and inexact and they are probably not relevant for your results. It could induce unnecessary criticism on your study. Unless recommended by your expert in statistics, I would not mention it.
  10. In line 197, “The range of eGFR reduction was 68-88mL/min/1.73m2” but I think that this is the range of eGFR at last evaluation, not the reduction. Please clarify it.
  11. Better use the symbol µg instead of mcg unless other indications from editors.
  12. I think that the need for additional tests (included US) in patients with CAKUT are not taken into account for cost analysis, please clarify it.
  13. Could text and numbers in Figures (specially in Figure 4) appear in a bigger size? I find it a bit difficult to read.

Kind regards and thank you again for this very relevant article.

Author Response

Reviewer 4

Thank you very much for your nice study. I consider it of a great merit taking into account the length of patients’ follow-up.

The objectives are relevant and results are clearly exposed. Most of minor changes are suggested rather than mandatory, or just additional explanations. Please consider them, if possible taking into account limits for extension of the article.

  1. In a relatively recent paper published in Pediatric Nephrology, some prognostic differences in CSFK have been suggested between renal agenesia and MCDK. Those patients are commonly grouped for studies on CSFK. This does not invalidate your study as proportion of MCDK in <2 and >2SDS is reflected in Table 1 (and no significant differences are found) and, besides, prognostic factors associated with poorer outcomes in MCDK are related to CAKUT and lack of compensatory hypertrophy, which have been considered. But maybe this fact would merit a comment.

Answer: we added these comments in the new version of the study, please see lines 385-392 of the new version of the manuscript.

  1. Age at last evaluation in clearly stated. Because these are relatively young adults, I think that utility of ERUSA in differentiating patients with good prognosis should be clearly stated in Summary as “In conclusion, only a RL>2SDS at 1–3months and 1year of age predicted good prognosis in YOUNG adulthood.” Anyway, you prudently recommend follow-up during adulthood, even in patients with early compensatory hypertrophy, which is reasonable.

Answer: we modified the text accordingly (please see line 44 of the new version of the manuscript).

  1. Do not use the term “urinary microalbumin” as microalbumin does not exist as a substance and KDIGO advices against the term “microalbuminuria”, use “urinary albumin” or “albuminuria”.

Answer: we modified the text accordingly (please see lines 141, 150, 224-225, 245, and Table 1 of the new version of the manuscript).

  1. If possible, different methods for creatinine measurements should be explained (probably because of changes in lab techniques across the years). Same for the use of original Schwartz formula.

Answer: in our laboratories the Jaffé method is used yet. Only for the last follow-up in adulthood, corresponding to the enrollment in this study, with a research purpose, the serum creatinine was measured by isotope dilution mass spectrometry. We better specified this concept in the new version of the manuscript. Please see lines 141-142 of the new version of the manuscript.

  1. Definitions for different CAKUT diagnosis are arguable but acceptable. Specifically talking about UPJO, I would rather use the term “compatible” rather than “positive” MAG3 scintigraphy, because it is a bit confusing. I assume you are talking about type 2 curves with a plateau in excretion, I do not know whether your urologist consider for surgery without derangement in relative kidney function. Type MAG3, not Mag3.

Answer: we modified the text accordingly (please see line 114 of the new version of the manuscript). Yes, I’m talking about curve with a plateau in excretion. Yes, our urologist considers for surgery the derangement in relative kidney function but in the case of congenital solitary kidney this is not applicable because we have a solitary functioning kidney and then we are not able to compare the relative kidney function between two kidneys. All over the manuscript we modified “Mag3” with “MAG3”.

  1. Expertise of radiologists and specialists in Nuclear Medicine in studies on pediatric patients should be stated, as well as how many of them perform this test (length measurements can vary from one observer to other). This would give readers an idea about the reproducibility on their own facilities, as accuracy could change with studies interpreted by non-expert specialists.

Answer: we added this information in the new version of the manuscript. Please see lines 127-131 of the new version of the manuscript.

  1. Time spent on procedures could be arguable (maybe overestimated in my personal opinion for lab procedures as determinations are performed at the same time for multiple samples) but I would accept it as it is well explained.

Answer: we made an estimation of the time spent considering also the time spent in the waiting room for the procedures. I recognize that the time is highly variable across cities, regions and nations but our aim is only to give an idea of the cost sparing applying ERUSA.

  1. Please type “Windows” (the operating system) with initial capital letter.

Answer: we modified the text accordingly (please see line 209 of the new version of the manuscript).

  1. Post-hoc power calculations can be tricky and inexact and they are probably not relevant for your results. It could induce unnecessary criticism on your study. Unless recommended by your expert in statistics, I would not mention it.

Answer: it has not been recommended by our expert in statistics, we deleted this paragraph according to your suggestions.

  1. In line 197, “The range of eGFR reduction was 68-88mL/min/1.73m2” but I think that this is the range of eGFR at last evaluation, not the reduction. Please clarify it.

Answer: the written English construction made this sentence difficult to understand. We modified the written English in order to make this sentence easier to follow (please see lines 222-226 of the new version of the manuscript.

  1. Better use the symbol µg instead of mcg unless other indications from editors.

Answer: we modified the text accordingly, please see lines 151, 226 and Table 1 of the new version of the manuscript.

  1. I think that the need for additional tests (included US) in patients with CAKUT are not taken into account for cost analysis, please clarify it.

Answer: we included also renal ultrasound in the cost analysis. We better specified what procedures we included in the new version of the manuscript (please see lines 182-183).

  1. Could text and numbers in Figures (specially in Figure 4) appear in a bigger size? I find it a bit difficult to read.

Answer: we increased the resolution and the size of the Figure 4.

Round 2

Reviewer 2 Report

Dear authors,

congratulations for this improved version of your work. 

All of my best regards.